# A Study on the Behavior Patterns of Liquid Aerosols Using Disinfectant Chloromethylisothiazolinone/Methylisothiazolinone Solution

**DOI:** 10.3390/molecules26195725

**Published:** 2021-09-22

**Authors:** Yong-Hyun Kim, Mi-Kyung Song, Kyuhong Lee

**Affiliations:** 1Department of Environmental Environment, Sangji University, Wonju 26339, Korea; 2Inhalation Toxicology Center for Airborne Risk Factor, Korea Institute of Toxicology, Jeongeup 56212, Korea; mikyung.song@kitox.re.kr; 3Humidifier Disinfectant Health Center, Korea Institute of Toxicology, Jeongeup 56212, Korea; 4Human and Environmental Toxicology, University of Science & Technology (UST), Daejeon 34113, Korea

**Keywords:** methylisothiazolinone, chloromethylisothiazolinone, liquid aerosols, breakthrough test

## Abstract

This study evaluates the behavioral characteristics of components (methylisothiazolinone (MIT) and chloromethylisothiazolinone (CMIT)) contained in disinfectant solutions when they convert to liquid aerosols. The analytical method for MIT and CMIT quantitation was established and optimized using sorbent tube/thermal desorber-gas chromatography-mass spectrometry system; their behavioral characteristics are discussed using the quantitative results of these aerosols under different liquid aerosol generation conditions. MIT and CMIT showed different behavioral characteristics depending on the aerosol mass concentration and sampling time (sampling volume). When the disinfectant solution was initially aerosolized, MIT and CMIT were primarily collected on glass filter (MIT = 91.8 ± 10.6% and CMIT = 90.6 ± 5.18%), although when the generation and filter sampling volumes of the aerosols increased to 30 L, the relative proportions collected on the filter decreased (MIT = 79.0 ± 12.0% and CMIT = 39.7 ± 8.35%). Although MIT and CMIT had relatively high vapor pressure, in liquid aerosolized state, they primarily accumulated on the filter and exhibited particulate behavior. Their relative proportions in the aerosol were different from those in disinfectant solution. In the aerosol with mass concentration of ≤5 mg m^−3^, the relative proportion deviations of MIT and CMIT were large; when the mass concentration of the aerosol increased, their relative proportions constantly converged at a lower level than those in the disinfectant solution. Hence, it can be concluded that the behavioral characteristics and relative proportions need to be considered to perform the quantitative analysis of the liquid aerosols and evaluate various toxic effects using the quantitative data.

## 1. Introduction

The number of cases of health damage caused by household chemical products has increased with an increase in the use (mainly, incorrect use) of household chemical products [1,2,3]. Hence, it is necessary to quantify harmful chemicals that the human body can be exposed to while using household chemical products and to evaluate their risks to human health [4,5,6,7]. Risk assessment requires an initial exposure assessment that is conducted based on the collection and analysis of target materials [8,9]. Household chemical products in liquid or solid state can be converted into aerosols during their usage [10,11]. Since the collection efficiency of aerosols in air varies depending on the types of sampling approaches, an effective sampling approach should be used that considers the behavioral characteristics of the aerosols [12]. Filters (i.e., glass fiber filter) can effectively collect aerosols that are in the form of fine solid particles; hence, the filter sampling approach is preferred for aerosol particles [13,14,15]. However, for liquid aerosols (liquid droplets), the filter collection efficiency differs based on the sampling conditions and physicochemical properties of the chemicals contained in the liquid aerosols [12,16,17]. For example, when liquid aerosols are continuously collected using a filter, the particles compromise the integrity of the filter and can break through [12,18]. The filter breakthrough characteristics of the liquid aerosol are different depending on the aerosol sampling flow rate and volume [12,14,18,19,20]. Additionally, the possibility of a change in the relative proportion of contained chemicals, when the solution becomes a liquid aerosol, cannot be excluded (due to the phase change, convection effect, and other factors). Therefore, to accurately quantify the liquid aerosols and the chemicals contained therein, the behavioral characteristic assessment under liquid aerosol generation conditions is required.

Methylisothiazolinone (MIT) biocides are used to control microbial growth in a wide range of personal care products, including lotions, sanitary wipes, shampoos, cosmetics, and humidifier disinfectants [12,21,22,23,24,25,26]. MIT and chloromethylisothiazolinone (CMIT) are the two most commonly used chemicals [12,21,23]. In Korea, humidifier disinfectant products containing CMIT/MIT were prevalent in the 2000s, and consumers used these products in humidifiers [12,27,28,29]. Studies are being actively conducted to determine whether CMIT/MIT aerosols generated through humidifiers cause inhalation toxicity. CMIT/MIT aerosols above a certain mass concentration level have been known to cause the inflammation of the respiratory tract [30]. For this reason, studies have been conducted to estimate the airborne concentrations of MIT and CMIT the users are exposed to by aerosolizing the liquid chemical products containing MIT and CMIT through a humidifier [31,32]. Park et al. [31] estimated that the airborne concentrations of MIT and CMIT averaged 0.34 ± 0.05 μg m^−3^ and 0.90 ± 0.14 μg m^−3^, respectively, when the liquid chemical products were added to the humidifier (recommended usage). However, data on the behavioral characteristics of CMIT/MIT aerosols that can be used to determine the process by which CMIT/MIT aerosols reach the respiratory tract are lacking. Due to the lack of data on the behavioral characteristics of liquid aerosols, it is difficult to quantify them by directly collecting and analyzing the chemicals contained in the aerosols. Generally, the concentration of the chemicals contained in the liquid aerosols is calculated by measuring the mass concentration of the liquid aerosol (filter sampling and measuring the weight of filter collecting the liquid aerosols) and applying the relative proportions of the chemicals contained in the solution before being aerosolized [33,34]. As the relative proportions of the chemicals contained in the solution and liquid aerosols vary depending on environmental conditions, direct sampling and analysis along with environmental conditions are required for accurate quantitative analysis of the chemicals contained in the liquid aerosols.

In this study, the behavioral characteristics of liquid aerosols for accurate quantitation were evaluated using aerosolizing liquid samples containing MIT and CMIT under various environmental conditions (i.e., sampling volume, mass concentration, etc.). The behavioral characteristics, in the present study, refer to the movement that affects the quantitative concentration of the components in the aerosols (i.e., from the point of view of sampling, aerosols with particulate behavior can be collected by filter and gaseous behavior requires adsorption or absorption sampling, yet another behavior is the relative mass proportion of each component in the aerosol mixture, etc.). By measuring the mass collected in the filter for the different sampling volumes and mass concentrations of the liquid aerosols, we determined whether the liquid aerosols behave as particulate matter or gas (the filter breakthrough test for MIT and CMIT in the aerosols). Additionally, the relative mass proportions of MIT and CMIT in the aerosols were assessed with different aerosol mass concentrations. Through this study, the behavioral characteristics of the CMIT/MIT aerosols were identified. Furthermore, we intend to apply the behavioral characteristics assessment for accurate quantitative analysis of various liquid aerosols. The establishment of a quantitative method for analyzing liquid aerosols can be a useful step in the inhalation toxicity testing of various liquid chemical products.

## 2. Materials and Methods

### 2.1. Experimental Scheme

The CMIT/MIT aerosols were generated using a spray method with a mist generator (NB2N, Sibata, Soka, Saitama, Japan) (Table 1). The aerosols were collected on the glass fiber filter (HF00025C, HYUNDAI Micro, Seoul, Korea) and sorbent tube (packed with Carbopack C, Supelco, Bellefonte, PA, USA); the CMIT/MIT were analyzed using thermal desorber (TD, TD20, Shimadzu, Kyoto, Japan)-gas chromatography (GC, GC2010, Shimadzu, Japan)-mass spectrometry (MS, GCMS-QP2010 ultra, Shimadzu, Japan) (TD-GC-MS) system. The behavioral properties of the CMIT/MIT aerosols were assessed through two experimental stages (Exp stage 1 and Exp stage 2). In Exp stage 1, the filter breakthrough test was conducted for different mass concentrations and sampling volumes of the CMIT/MIT aerosols. The boundary between particulate and gaseous behavior of the CMIT/MIT aerosols was determined depending on the occurrence of filter breakthrough. In Exp stage 2, the relative proportions of MIT and CMIT in the aerosols were assessed with different aerosol mass concentrations (Figure 1 and Table 2).

### 2.2. Preparation of Working Standards and Solutions

The working standards (WSs) for quantitative analysis of MIT and CMIT in the aerosols and sample solutions (SSs) for generation of CMIT/MIT aerosols were prepared (the details of the same are included in Appendix A). The primary standard was purchased at purities of 0.43% MIT and 1.09% CMIT (Sigma-Aldrich, St. Louis, MO, USA). WSs were prepared by diluting the primary standard in distilled water to eight different concentrations (47.8–4306 ng µL^−1^ (MIT) and 121–10,916 ng µL^−1^ (CMIT)). The primary solution for preparing SSs was purchased in bulk (640 kg) from Dow Chemical Group (product number: 00010039070, Midland, MI, USA) for CMIT/MIT aerosol generation with purities of 0.368% MIT and 1.124% CMIT. The SSs at three different concentrations were prepared by gravimetric dilution of the primary solution in distilled water to generate CMIT/MIT aerosols with concentrations ranging from 0.20 to 60 mg m^−^^3^ (23.8–71.5 ng µL^−1^ (MIT) and 72.8–218 ng µL^−1^ (CMIT)).

### 2.3. Instrumental Setup for Analysis of MIT and CMIT

The glass fiber filter and sorbent tube loading the CMIT/MIT aerosol samples were thermally desorbed at 250 °C (for 5 min) in a reverse flow of 100 mL min^−1^ of helium carrier gas (He > 99.999%). The desorbed analytes were swept into the cold trap (held at 0 °C) in the stream of carrier gas. The cold trap itself was then rapidly desorbed (280 °C for 5 min) in a reverse flow of carrier gas to transfer (inject) the MIT and CMIT into the HP-5 GC capillary column (diameter: 0.32 mm, length: 30 m, thickness: 0.25 µm; Agilent J&W, Santa Clara, CA, USA). The transfer/injection of analytes from the TD cold trap to the GC column was carried out in split, with 1 and 20 mL min^−1^ passing to the column and split vent, respectively. To optimize the analysis of MIT and CMIT, the oven temperature was initially set at 60 °C, then ramped up to 140 °C at a rate of 10 °C min^−1^ and to 280 °C at a rate of 40 °C min^−1^; this temperature was maintained for 3.5 min (total run time of 16 min).

The analytes separated by the GC system were detected by the MS system. The interface and ion source temperatures were set to 280 °C. Analytes were initially analyzed in the total ion chromatographic (TIC) mode over a mass range of 50 to 400 *m*/*z*. The extracted ion chromatographic (EIC) mode was subsequently applied to minimize interference and maximize recovery by using the significant ions that were identified from the main spectra of each analyte (115 and 149 *m*/*z* for MIT and CMIT, respectively) (Table 1). Detailed instrumental conditions are presented in Table 3 [12].

### 2.4. Experimental Approaches

#### 2.4.1. Working Standards

In this study, a total of four types of calibration curves were obtained using the WSs depending on the CMIT/MIT concentration ranges and the aerosol sampler types. To obtain the calibration curves, the WSs were directly injected into the filter or sorbent tube and analyzed. The WSs were divided into low (47.8–478 ng µL^−1^ (MIT) and 121–1213 ng µL^−1^ (CMIT)) and high (431–4306 ng µL^−1^ (MIT) and 1092–10,916 ng µL^−1^ (CMIT)) concentration groups, respectively ((1) MIT and CMIT calibration curves in low concentration range using filter, (2) in high concentration range using filter, (3) in low concentration range using sorbent tube, and (4) in high concentration range using sorbent tube) (Figure 2).

Two types of sampling tubes were prepared ((1) filter tube: by inserting the glass fiber filter in empty quartz holders; and (2) sorbent tube: by packing 100 mg of Carbopack X in empty quartz holders)). The inlet of the tubes was connected to a Tenax tube packed with Tenax TA (100 mg) for filtered air; the outlet tube was connected to a vacuum pump interfaced with the mass flow controller (ΣMP-30, Sibata, Japan). In the case of the filter tube with the glass fiber filter, 1 μL of the WSs were spiked into the filter in the empty quartz holders, while filtered air was constantly supplied into the quartz tube at 200 mL min^−1^ for 5 min. In the sorbent tube, the 1 μL of WSs were injected into the sorbent tube, while filtered air was pumped through the tube from the reservoir at a rate of 200 mL min^−1^ for 5 min. By supplying air to the filter and adsorbent-loaded WSs, the interference effect on MIT and CMIT by the WS solvent was minimized [35,36,37]. The filter tube and sorbent tube-loaded samples were then analyzed using the TD-GC-MS system.

#### 2.4.2. CMIT/MIT Aerosol

The SS (500 mL) containing MIT and CMIT was aerosolized using the mist generator. The aerosols were generated at room temperature (25 °C ± 3) for 2 h and the mass concentration of the aerosol was controlled by suppling external air during the aerosolization process. Temperature variables that could affect the aerosol behavior were excluded by fixing the chamber temperature at room temperature when generating the aerosols. The CMIT/MIT aerosol samples were collected by the glass filter (primary sampler) and sorbent tube packed with Carbopack X (secondary sampler). The inlet of the glass filter holder was connected to the outlet of the mist generator and the outlet of the filter holder was connected to the inlet of the sorbent tube. The outlet of the sorbent tube was connected to a vacuum pump. The transfer of the CMIT/MIT aerosols generated from the mist generator into the glass filter and sorbent tube was initiated at a fixed flow rate of 0.5 L min^−1^. Weight from the glass filter loaded with the CMIT/MIT aerosols was measured before and after aerosol sampling (LE225D, Sartorius, Göttingen, Germany). The aerosol mass concentrations were then calculated using the sampling weight and volume data, and this was followed by inserting the glass filters containing the CMIT/MIT aerosols into the empty quartz trap. The quartz tube containing the glass filter and the sorbent tube were then thermally desorbed and analyzed using the TD-GC-MS system. In Exp stage 1, the filter breakthrough test was conducted by comparing the concentrations of MIT and CMIT (collected on) between glass fiber filter and the sorbent tube. In Exp stage 2, the relative proportions of MIT and CMIT with different aerosol mass concentrations were assessed by summing the concentrations of MIT (or CMIT) collected on the filter and the sorbent tube. The chromatograms of MIT and CMIT with different aerosol sampling volumes are presented in Figure 3.

## 3. Results and Discussion

### 3.1. Calibration and QA Data

The four types of calibration results obtained using the working standards (WSs) are presented in Figure 2. The response factors (RF, ng^−1^) of MIT and CMIT were similar, regardless of the sampler types. All calibration curves for MIT and CMIT exhibited good linearity of above 0.99 (mean 0.9936 ± 0.0023, *n* = 4). The reproducibility of experimental data was assessed in terms of relative standard error (RSE, %) using triplicate analyses of WS-3. The RSE values of MIT and CMIT were fairly stable, i.e., <2% (mean 1.20 ± 0.53%, *n* = 4). The MIT and CMIT detection limits of the TD-GC-MS system were estimated by limit of detection (LOD). The LOD values were determined from three times the standard deviation of background noise (*n* = 7), yielding values of <0.4 ng m^−3^ (calculated using the calibration data obtained from low concentration WSs (WS-1 to WS-4)) and <3.6 ng m^−3^ (calculated using the calibration data obtained from high concentration WSs (WS-5 to WS-8). The detection limit levels of MIT and CMIT are low enough to detect the aerosolized MIT and CMIT in this study. Hence, the sampling and analytical method for quantitation of CMIT/MIT is reliable enough to achieve high precision and sensitivity.

### 3.2. MIT and CMIT Behavior Patterns with Different Sampling Volumes and Mass Concentrations of CMIT/MIT Aerosols (Exp Stage 1)

The CMIT/MIT aerosols were collected on the glass fiber filter, following which, the MIT and CMIT particles that passed through the filter were continuously collected by sorbent tubes. The CMIT/MIT aerosols in concentrations ranging from 0.2 to 16.7 mg m^−3^ were sampled with different sampling volumes (3, 6, 10, 15, and 30 L) (Figure 4). Assuming that the sum of the concentrations of the sample collected on the filter and the sorbent tube was 100%, each concentration of the samples collected on the filter and the sorbent was calculated as the relative concentrations, as shown in (Equation (1)).
(1)$$Relative\;concentration\;\left(\%\right)=\frac{Concentration\;\left(sample\;collectd\;on\;filter\;or\;ST\right)}{Concentration\;\left(sample\;collectd\;on\;filter\;and\;ST\right)}\times 100$$

To assess the change in the relative concentrations of MIT and CMIT with different aerosol sampling volumes and mass concentrations, the linear regression analysis was conducted and the slope and R^2^ for the linear regression line were calculated (Figure 5). In MIT, when the sampling volume of CMIT/MIT aerosols was ≤15 L, the relative concentration of MIT at the filter averaged 92.0 ± 8.22%. Although the relative concentration of the filter was reduced to an average of 79.0 ± 12.0% at a sampling volume of 30 L, it was confirmed that MIT containing aerosols were primarily collected by the filter with relative concentration of >90% up to the 15 L aerosol sampling volume. Linear regression analysis exhibited a slope of −0.5107, and the relative concentration of MIT at the filter decreased as the aerosol sampling volume increased; however, the R^2^ value was as low as 0.2239. The relative concentrations of MIT collected on the filter increased with the increasing CMIT/MIT aerosol concentrations. The relative concentration of MIT collected on the filter was the mean value of 76.1 ± 9.44% at an aerosol concentration of <2 mg m^−3^. When the concentration of CMIT/MIT aerosol increased to 10 mg m^−3^, the relative concentration of MIT collected on the filter increased to 95.9 ± 4.82%. The relative concentration of MIT at the filter increased systematically with increasing CMIT/MIT aerosol concentration (relative concentrations of MIT collected on the filter ± SD (aerosol mass concentration range) = 76.1 ± 9.44% (0.2–2 mg m^−3^), 88.7 ± 6.26 (2–5 mg m^−3^), 95.9 ± 4.82% (5–10 mg m^−3^), and 94.9 ± 5.85% (10–16.7 mg m^−3^)). When the mass concentration of the aerosol was >10 mg m^−3^, the relative concentration of MIT at the filter was >90%. Most of the MIT contained in the aerosols were collected by the filter at >10 mg m^−3^ of aerosol mass concentration, regardless of the aerosol sampling volume. As shown in Figure 5c, the slope of the linear regression line (relative concentration of MIT collected on the filter vs. aerosol mass concentration) was 1.297 and the R^2^ value was 0.3744. If the linear regression analysis was recalculated with only the aerosol mass concentration data corresponding to <10 mg m^−3^, then both slope and R^2^ values increased to 2.9837 and 0.5152, respectively.

In the case of CMIT, the relative concentration of the filter decreased sharply with increasing aerosol sampling volume. Although the relative concentration of CMIT in the filter was relatively high (90.6 ± 5.18%) at 3 L aerosol sampling volume, it dropped substantially to 39.7 ± 8.35% at 30 L aerosol sampling volume (relative concentrations of CMIT collected on the filter ± SD (aerosol sampling volume) = 90.6 ± 5.18% (3 L), 83.5 ± 5.27 (6 L), 65.9 ± 7.74% (10 L), 56.7 ± 4.09% (15 L), and 39.7 ± 8.35% (30 L)). A slope of −1.8467 was obtained through linear regression analysis between the relative concentration of CMIT collected on the filter and the aerosol sampling volume, which was three times lower than that of MIT. Additionally, the R^2^ value of the regression line for CMIT (relative concentration of the filter vs. aerosol sampling volume) was significantly high at 0.8268. Contrarily, CMIT did not display a significant change in the relative concentration of the filter with the increase in mass concentration of CMIT/MIT aerosols. As shown in Figure 5d, the relative concentration of CMIT collected on the filter showed the tendency to slightly increase as the aerosol mass concentration increased (slope of the aerosol mass concentration (mg m^−3^) vs. the relative concentration of CMIT collected on the filter (%) = 1.0091). However, the relationship between the relative concentration of CMIT at the filter and aerosol mass concentration showed a very low correlation with R^2^ = 0.064. Relative standard deviation (RSD) values for the relative concentrations of CMIT collected on the filter were calculated by dividing groups with the aerosol mass concentration levels; the average was significantly high at >28.0%. (mean relative concentration of CMIT collected on the filter ± SD (aerosol mass concentration range) 58.2 ± 22.2% (0.2–2 mg m^−3^), 61.3 ± 16.1 (2–5 mg m^−3^), 77.6 ± 18.4 (5–10 mg m^−3^), and 68.1 ± 16.1 (0.2–2 mg m^−3^)). Specifically, it is considered that the mass concentration of the aerosol did not have a substantial influence on the behavior of CMIT.

To learn more about the interactive relationships between different variables, the relative concentration of MIT and CMIT collected on the filter was evaluated in relation to the aerosol sampling volumes or aerosol mass concentrations using one-way analysis of variance (ANOVA) (Table 4). In both MIT and CMIT, the relative concentration patterns with different aerosol sampling volumes were statistically significant with *p*-values < 0.05. In particular, CMIT exhibited a considerably high statistical significance, with a *p*-value of 8.60 × 10^−20^. To elaborate, in MIT, only the 30 L aerosol sampling volume group displayed a statistically significant difference from other sampling volume groups (mean *p*-value ± SD = 1.97 × 10^−2^ ± 9.26 × 10^−3^). There was no statistically significant difference in the relative concentration of MIT collected on the filter among the aerosol sampling volumes of 3, 6, 10, and 15 L (mean *p*-value ± SD = 0.75 ± 0.16). In contrast to MIT, CMIT exhibited a statistically significant difference in all relative concentrations of the filter with the different aerosol sampling volumes (mean *p*-value ± SD = 1.68 × 10^−3^ ± 3.67 × 10^−3^). The statistical significance between the relative concentrations of MIT and CMIT in the filter and the aerosol mass concentration was opposite to that of the aerosol sampling volume. MIT had high statistical significance between the relative concentration of filter and aerosol mass concentration with *p*-value of 1.81 × 10^−8^. In the case of CMIT, there was no statistical significance between the relative concentration of the filter and the aerosol mass concentration (*p*-value = 0.0616). Specifically, MIT was statistically significant with *p*-values < 0.05 except for the ANOVA analysis results between aerosol mass concentrations of the groups corresponding to 5–10 mg m^−3^ and 10–16.7 mg m^−3^ (mean *p*-value ± SD = 8.57 × 10^−3^ ± 1.46 × 10^−2^). CMIT recorded a comparatively low mean *p*-value of 0.034 ± 0.012 in the aerosol mass concentration group corresponding to the 5–10 mg m^−3^ range, although there was no significant statistical difference in the relative concentration of the filter with different aerosol mass concentrations (mean *p*-value ± SD = 0.381 ± 0.240).

Sampling volume and aerosol concentrations are important variables that determine the particulate and gaseous behavior of aerosol components. In this study, we confirmed that the increase in the sampling volume and aerosol concentration induced a characteristic change from particulate behavior to gaseous behavior in aerosols, but the extent of conversion varied depending on the type of aerosol. However, further research is necessary to confirm such variations in the behavioral patterns that exist among different aerosol types.

### 3.3. Relative Proportion Changes of MIT and CMIT at Different Aerosol Concentrations (Exp Stage 2)

The relative proportions of MIT and CMIT in the aerosol were calculated as follows (Equation (2)):(2)$$Relative\;proportion\;\left(\%\right)=\frac{Mass\;of\;MIT\;or\;CMIT\;\left(collectd\;on\;filter\;and\;ST\right)}{Aerosol\;Mass}\times 100$$

The masses of MIT and CMIT were calculated by analyzing the CMIT/MIT aerosols collected by filter and ST using the TD-GC-MS system. The aerosol mass was calculated by measuring the weight of the filter loading the aerosol using a scale (detailed in Section 2.4.2). 

The relative proportions of MIT and CMIT contained in the sample solution (SS) were 0.368% and 1.124%, respectively. During filter collection, the CMIT/MIT aerosol particles (solids) except water were collected, and the ideal relative proportion (relative mass proportions in the particles excluding water in the SS) of MIT and CMIT in the aerosol was 1.5275% and 4.67%, respectively (Figure 6).

Higher relative proportions of MIT and CMIT were observed for lower aerosol mass concentrations. When the aerosol mass concentration was <5 mg m^−3^, the relative proportions of MIT and CMIT in the aerosol were relatively high at 1.65 ± 0.43% and 2.77 ± 0.53%, respectively (*n* = 78). In the case of MIT, the mean relative proportion was even higher than the ideal relative proportion. However, the reproducibility of the relative proportions of MIT and CMIT was low, with high RSD values of 26.0% and 19.2%, respectively, at <5 mg m^−3^ aerosol mass concentrations. Additionally, the difference between the maximum and minimum values of the relative proportions of MIT and CMIT at <5 mg m^−3^ aerosol mass concentration was approximately 3.5 and 2.5 times, respectively. Hence, it was confirmed that the reproducibility of the relative proportions of MIT and CMIT was poor when the CMIT/MIT aerosol had relatively low mass concentrations, such as at concentration of <5 mg m^−3^. However, as the mass concentrations of the CMIT/MIT aerosol increased to >5 mg m^−3^, the relative proportions of MIT and CMIT converged to certain levels comparatively lower than the ideal relative proportions. For aerosol mass concentrations in the range 5–60 mg m^−3^, MIT and CMIT had relative proportions of mean 0.96 ± 0.18% and mean 1.87 ± 0.29%, respectively. Their RSD values were relatively low at <20%. For aerosol mass concentrations in the range of 30–60 mg m^−3^, the mean relative proportions of MIT and CMIT recorded were 0.90 ± 0.11 and 1.82 ± 0.18%, respectively, and the RSD values of MIT and CMIT were even lower at <12%. Specifically, at <5 mg m^−3^ aerosol mass concentration, the mean relative proportions of MIT and CMIT were high with poor repeatability. In contrast, when the aerosol mass concentration increases to >5 mg m^−3^, the relative proportions of MIT and CMIT converged to the constant levels that were comparatively lower than the relative proportions in the SS. Therefore, it was confirmed that the relative proportions of organic compounds existed in certain levels in the solution, and fluctuated when the solution was aerosolized; however, the observed fluctuations can vary depending on the aerosol mass concentration levels and the types of organic compounds.

## 4. Conclusions

We use diverse liquid household chemical products containing organic compounds, and our exposure levels to these organic compounds varies depending on the usage of the household chemical products. Liquid household chemical products can be aerosolized depending on different use conditions, and the aerosol behavior affects the determination of the quantity of their exposure to humans. In this study, liquid disinfectant products containing CMIT/MIT were aerosolized and the behavioral characteristics of the liquid aerosols were assessed by collecting and analyzing MIT and CMIT levels in the aerosol samples. A filter breakthrough test was conducted for MIT and CMIT samples with different aerosol mass concentrations and generating volume (sampling volume). Additionally, the change in the relative proportions of MIT and CMIT in the aerosols was evaluated with different CMIT/MIT aerosol mass concentrations.

The aerosol mass concentrations were an important variable in determining the filter breakthrough of MIT in the aerosol, and the aerosol sampling volume was the main condition in determining the filter breakthrough of CMIT. In MIT, as the mass concentration of the aerosol decreased, the mass collected in the filter decreased. When the aerosol mass concentrations were lowered from 16.7 to 0.2 mg m^−3^, the relative concentration of MIT collected in the filter decreased from 99.5% to 57.7%. In CMIT, the mass collected in the filter significantly decreased with increasing aerosol sampling volume. The relative concentration of CMIT collected in the filter was close to 100% (96.6%) in the aerosol sampling 3 L, although when the aerosol sampling volume was 30 L, it dropped to a minimum of 27.2%. At a relatively low level of the aerosol mass concentration (<5 mg m^−3^), both MIT and CMIT exhibited large deviations in relative proportions (approximately 3.5 times), and when the aerosol mass concentration increased, both MIT and CMIT had constant relative proportion values. The relative proportions of MIT and CMIT re-formed in the aerosol at above a certain mass concentration was comparatively lower than those in the solution. 

This study demonstrated that filter breakthrough occurred under different conditions for MIT and CMIT in the aerosol, and that the relative proportions of MIT and CMIT also differed with the aerosol mass concentrations. Hence, for accurate quantitative evaluation of components in aerosols, the sampling and analysis should be conducted considering the conditions of aerosol mass concentration and generating volume (sampling volume). Accordingly, the research methods and findings presented in this study can be used effectively to increase the accuracy of aerosol quantitative analysis. This is an essential step in testing various liquid chemicals for safety. As part of a follow-up study, we intend to evaluate the exposure levels of chemicals in liquid aerosols under different practical conditions of its use.

## Figures and Tables

**Figure 1 molecules-26-05725-f001:**
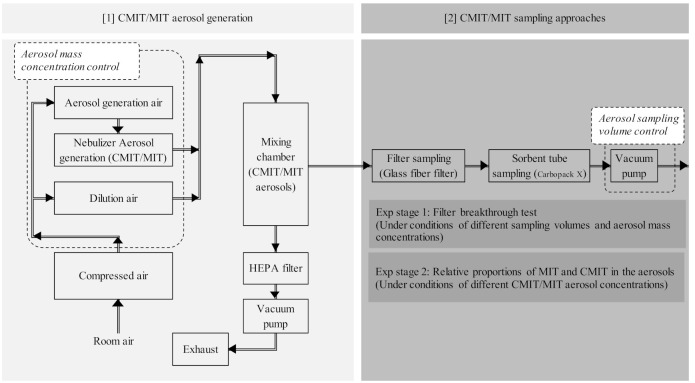
Flow chart for CMIT/MIT aerosol generation and sampling approaches.

**Figure 2 molecules-26-05725-f002:**
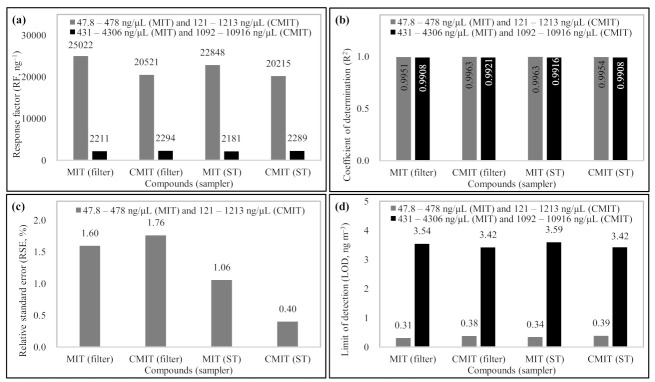
Basic calibration and QA results using working standards: (**a**) response factor (RF, ng^−1^), (**b**) coefficient of determination (R^2^), (**c**) relative standard error (RSE, %), and (**d**) limit of detection (LOD, ng m^−3^).

**Figure 3 molecules-26-05725-f003:**
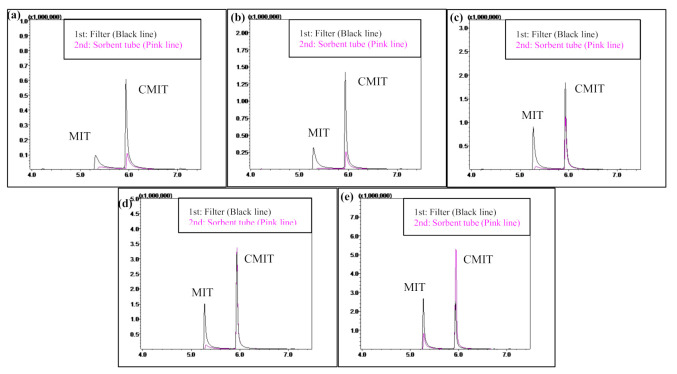
Chromatograms of the MIT and CMIT with different CMIT/MIT aerosol sampling volumes: (**a**) 3 L, (**b**) 6 L, (**c**) 10 L, (**d**) 15 L, and (**e**) 30 L; X-axis = retention time (min) and Y-axis = peak area.

**Figure 4 molecules-26-05725-f004:**
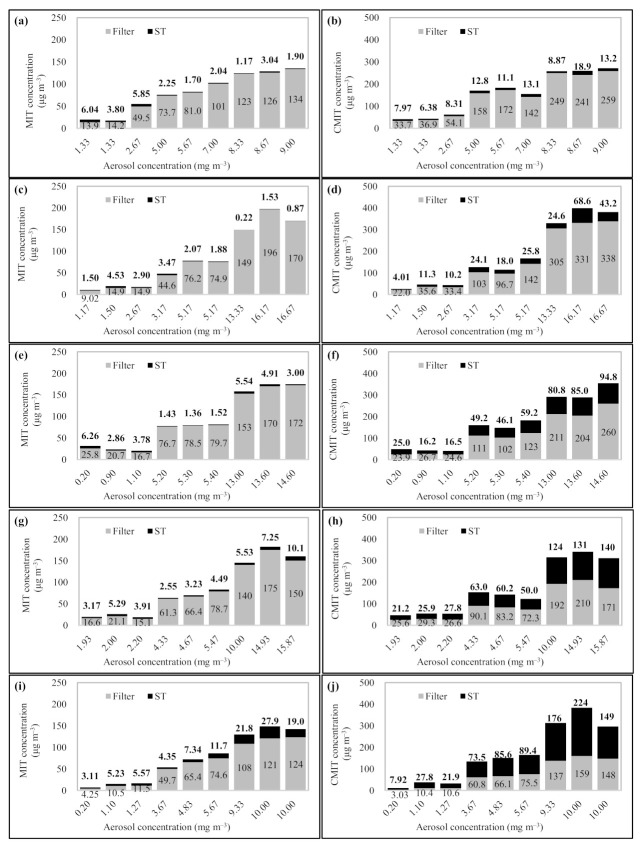
Results of the breakthrough test of the glass fiber filter using the CMIT/MIT aerosol samples with different sampling volume and aerosol concentrations: (**a**) sampling volume 3 L (MIT), (**b**) sampling volume 3 L (CMIT), (**c**) sampling volume 6 L (MIT), (**d**) sampling volume 6 L (CMIT), (**e**) sampling volume 10 L (MIT), (**f**) sampling volume 10 L (CMIT), (**g**) sampling volume 15 L (MIT), (**h**) sampling volume 15 L (CMIT), (**i**) sampling volume 30 L (MIT), and (**j**) sampling volume 30 L (CMIT).

**Figure 5 molecules-26-05725-f005:**
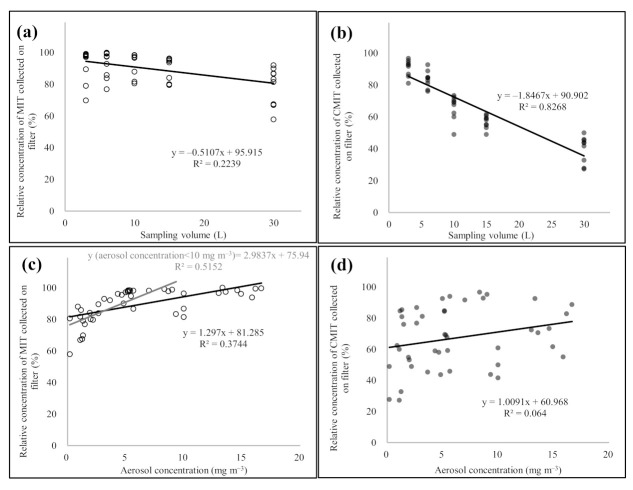
Scatter plot depicting the comparison of the relative concentrations of MIT and CMIT collected on the glass fiber filter with different sampling volumes and the CMIT/MIT aerosol concentrations. (**a**) MIT: relative concentration vs. sampling volume, (**b**) CMIT: relative concentration vs. sampling volume, (**c**) MIT: relative concentration vs. aerosol concentration, and (**d**) CMIT: relative concentration vs. aerosol concentration.

**Figure 6 molecules-26-05725-f006:**
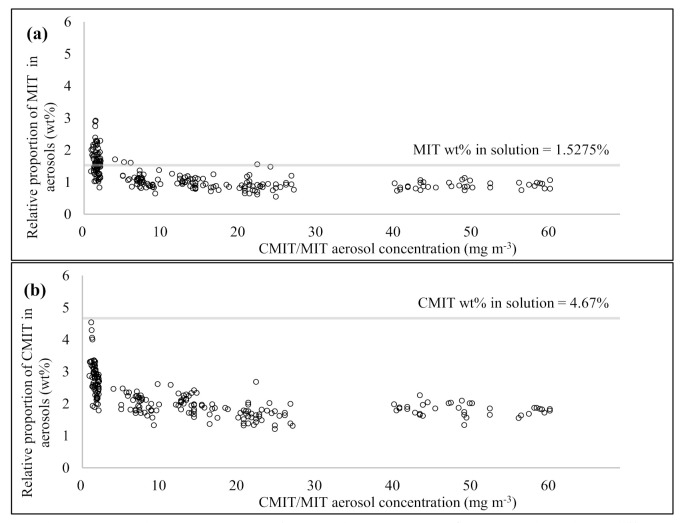
Scatter plot showing relative proportions of MIT and CMIT in CMIT/MIT aerosols with the different aerosol mass concentrations. (**a**) MIT and (**b**) CMIT.

**Table 1 molecules-26-05725-t001:** Basic information of target compounds in this study.

Full name:	Methylisothiazolinone (2-Methyl-4-isothiazolin-3-one)	Chloromethylisothiazolinone (5-chloro-2-methyl-4-isothiazolin-3-one)
Short name:	MIT	CMIT
MW (g mol^−1^):	115.1	149.59
Density (g mL^−3^):	1.293	1.02
Boiling point (°C):	182.8	200.2
Formula:	C_4_H_5_NOS	C_4_H_4_ClNOS
CAS number:	2682-20-4	26172-55-4
Main spectra ^1^ (*m*/*z*):	115	149
Chemical structure:	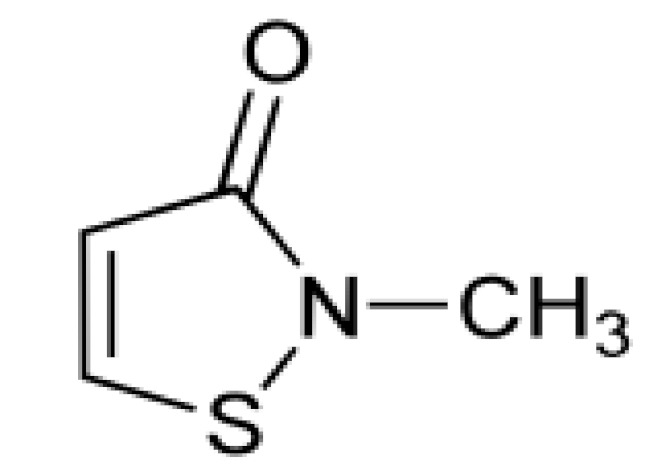	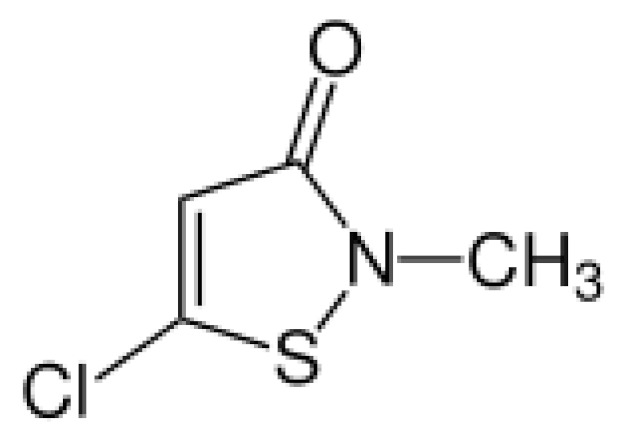

^1^ Mass spectra selected for the EIC-base analysis.

**Table 2 molecules-26-05725-t002:** Experimental scheme for the testing of the CMIT/MIT aerosol behavior patterns.

A. Experimental Stage Information			
Order	Exp Code	Contents	
1	Exp stage 1	Filter breakthrough test of MIT and CMIT in the aerosols	
2	Exp stage 2	Assessment of the relative proportions of MIT and CMIT in the aerosols with different aerosol mass concentrations	
**B. Information of the Breakthrough Test of the Glass Fiber Filter Using the CMIT/MIT Aerosol Samples (Exp Stage 1)**			
**Order**		**Preliminary test**	**Main test**
1	Sampling flow rate:	0.5 L min^−^^1^	0.5 L min^−^^1^
2	Sampling volume:	65 L	3, 6, 10, 15, and 30 L
3	Primary sampler:	Glass fiber filter ^1^	Glass fiber filter ^1^
4	Secondary sampler:	Sorbent tube ^2^	Sorbent tube ^2^
5	Tertiary sampler:	Sorbent tube ^2^	−
6	Aerosol concentration ^3^:	1.82 mg m^−^^3^	0.20–16.7 mg m^−^^3^
7	Sample number:	3	45
**C. Information of the Relative Proportion Test for MIT and CMIT in the Aerosol Samples (Exp Stage 2)**			
1	Sampling flow rate:	0.5 L min^−^^1^	
2	Primary sampler:	Glass fiber filter ^1^	
3	Secondary sampler:	Sorbent tube ^2^	
4	Aerosol concentration ^3^:	0.9–60 mg m^−^^3^	
5	Sample number:	225	

^1^ GF-C (pore size: 1.2 μm, Hyndai Micro, Seoul, Korea). ^2^ The sorbent tubes were packed with Carbopack × 200 mg (40/60 mesh, Sigma-Aldrich, Burlington, MA, USA). ^3^ Aerosol samples were generated from the sample solution (SS-1, SS-2, and SS-3) using the mist generator and collected by each sampler through the aerosol mixing chamber.

**Table 3 molecules-26-05725-t003:** Operational conditions for analysis of MIT and CMIT in aerosols.

A. Pretreatment: Thermal Desorber (Model: Shimadzu, Kyoto, Japan)			
a. Sampling Tube			
1. Trap tube:	Quartz (length: 90 mm, OD: 6.4 mm, and ID: 4.2 mm)		
2. Filter or Adsorbent:	Glass fiber filter or Carbopack × (200 mg)		
3. Desorption flow:	100 mL min^−1^ (to cold-trap)		
4. Desorption time:	5 min		
5. Desorption temp.:	250 °C		
**b. Cold-Trap**			
1. Trap tube:	Quartz coated by silcosteel (length: 100 mm, OD: 3.2 mm, and ID: 2 mm)		
2. Adsorbent:	Quartz wool and Tenax TA (volume ratio = 1:1)		
3. Adsorption temp.:	0 °C (from sampling tube)		
4. Desorption temp.:	280 °C (to GC)		
5. Desorption flow:	23 mL min^−1^		
6. Desorption time:	5 min		
**c. Carrier Gas Setting**			
1. Carrier gas:	Helium (> 99.999%)		
2. Initial gas flow:	1 mL min^−1^	(Constant gas flow)	
3. Split flow:	20 mL min^−1^ (Method A) and 200 mL min^−1^ (Method B)		
**d. Line and Interface Temp.: 280 °C**			
**B. Separation: Gas Chromatography (Model: GC-2010, Shimadzu, Kyoto, Japan)**			
**a. Column:**	HP-5 (Agilent J&W, California, USA)		
	(length: 30 m, diameter: 0.32 mm, and film thickness: 0.25 µm)		
**b. Oven Setting:**	60 °C (1 min) → 10 °C/min → 140 °C (0 min) → 40 °C/min → 280 °C (3.5 min)		
	(Total program time = 16 min)		
**C. Detection: Mass Spectrometry (Model: GCMS-QP2010 ultra, Shimadzu, Kyoto, Japan)**			
**a. Ionization Mode:**	EI (70 ev)	d. TIC scan range:	50~400 m/z
**b. Ion source Temp.:**	280 °C	e. Scan speed:	1250
**c. Interface Temp.:**	280 °C		

**Table 4 molecules-26-05725-t004:** Results of the statistical analysis of the relative concentrations of MIT and CMIT on the glass fiber filters with the different sampling volumes and the aerosol concentrations.

A. ANOVA Test (One-Way): Total				
Group Variable	Compound	Grouping		*p*-Value
Sampling volume	MIT	3 6, 10, 15, and 30 L		0.0149 *
	CMIT			8.60 × 10^−20^ **
Aerosol concentration	MIT	0.2−2, 2−5, 5−10, 10−16.7 mg m^−3^		1.81 × 10^−8^ **
	CMIT			0.0616
**B. ANOVA Test (One-Way): Individual**				
**a. Sampling Volume (L)**				
*p*-value	3	6	10	15
*(a) MIT*				
6	0.8825			
10	0.7989	0.9076		
15	0.7756	0.612	0.5075	
30	0.0294 *	0.0144 *	0.0097 **	0.0255 *
*(b) CMIT*				
6	0.0106 *			
10	5.99 × 10^−7^ **	3.79 × 10^−5^ **		
15	5.03 × 10^−11^ **	1.94 × 10^−9^ **	6.04 × 10^−3^ **	
30	4.51 × 10^−11^ **	4.61 × 10^−10^ **	3.54 × 10^−6 **^	5.04 × 10^−5^ **
**b. Aerosol concentration (mg m^−3^)**				
*p*-value	0.2−2	2−5	5−10	
*(a) MIT*				
2−5	2.920 × 10^−3^ **			
5−10	5.971 × 10^−7^ **	5.646 × 10^−3^ **		
10−16.7	1.631 × 10^−5^ **	0.0343 *	0.6702	
*(b) CMIT*				
2−5	0.7246			
5−10	0.0256 *	0.0427 *		
10–16.7	0.2442	0.3641	0.1912	
* *p*-value < 0.05				
** *p*-value < 0.01				

## Data Availability

On request, the computational data is available from Y.-H.K.

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
