# Peer review of "A Study on the Behavior Patterns of Liquid Aerosols Using Disinfectant Chloromethylisothiazolinone/Methylisothiazolinone Solution"

_molecules, 2021, doi:10.3390/molecules26195725_

Round 1
Reviewer 1 Report
Dear authors,
In уоur study, the behavioral characteristics of liquid aerosols for accurate quantitation were evaluated using aerosolizing liquid samples containing MIT and CMIT under various environmental conditions (i.e., sampling volume, mass concentration, etc.). By measuring the mass collected in the filter for the different sampling volumes and mass concentrations of the liquid aerosols, determined whether the liquid aerosols behave as particulate matter or gas (the filter breakthrough test for MIT and CMIT in the aerosols). Additionally, the relative mass proportions of MIT and CMIT in the aerosols were assessed with different aerosol mass concentrations. Through this study, the behavioral characteristics of the CMIT/MIT aerosols were identified.
I think the following should be done:
- 1. In the Abstract, change the following: In this manuscript, analytical methods for determining .......... have been done, presented, or optimized, and the obtained results have been discussed. Based on the obtained results, it can be concluded .........................
- 2. Based on the results presented in Table 5, using standards, it would be good to present a diagram.
- Table 6, which is quite cumbersome, needs to be corrected or the results presented in a diagram. Think about it
- You explain: The boundary between the particles and the gaseous behavior of the CMIT / MIT aerosol is determined depending on the occurrence of filter breakage. In Exp phase 2, the relative proportions of MIT and CMIT in aerosols were estimated with different aerosol mass concentrations (Figure 1 and Table 2). Next, you presented the results obtained by measuring the mass collected in the filter for different sampling volumes and mass concentrations of liquid aerosols. The phenomenon of gas-liquid balance, the influence of temperature and certain laws that determine it need to be clarified. Write at least a few sentences about it.
With respect
Author Response
Response to Reviewer 1 Comments
General Comments
In уоur study, the behavioral characteristics of liquid aerosols for accurate quantitation were evaluated using aerosolizing liquid samples containing MIT and CMIT under various environmental conditions (i.e., sampling volume, mass concentration, etc.). By measuring the mass collected in the filter for the different sampling volumes and mass concentrations of the liquid aerosols, determined whether the liquid aerosols behave as particulate matter or gas (the filter breakthrough test for MIT and CMIT in the aerosols). Additionally, the relative mass proportions of MIT and CMIT in the aerosols were assessed with different aerosol mass concentrations. Through this study, the behavioral characteristics of the CMIT/MIT aerosols were identified.
I think the following should be done:
Specific Comments
Point 1: In the Abstract, change the following: In this manuscript, analytical methods for determining .......... have been done, presented, or optimized, and the obtained results have been discussed. Based on the obtained results, it can be concluded .........................
Response 1: Thank you for the great comments. As you pointed out, some phrases were included in Abstract (page 1). Some phrases included are “In this manuscript, analytical methods, optimized, discussed, based on the obtained results, it can be concluded.”
Point 2: Based on the results presented in Table 5, using standards, it would be good to present a diagram.
Response 2: Thank you for the great comments. Table 5 is deleted and the data in Table 5 is presented in Figure 2 using bar graphs (page 6).
Point 3: Table 6, which is quite cumbersome, needs to be corrected or the results presented in a diagram. Think about it
Response 3: Thank you for the valuable suggestion. Table 6 is also deleted and the data in Table 6 is presented in Figure 4 using bar graphs (page 8).
Point 4: You explain: The boundary between the particles and the gaseous behavior of the CMIT/MIT aerosol is determined depending on the occurrence of filter breakage. In Exp phase 2, the relative proportions of MIT and CMIT in aerosols were estimated with different aerosol mass concentrations (Figure 1 and Table 2). Next, you presented the results obtained by measuring the mass collected in the filter for different sampling volumes and mass concentrations of liquid aerosols. The phenomenon of gas-liquid balance, the influence of temperature and certain laws that determine it need to be clarified. Write at least a few sentences about it.
Response 4: Thank you for the constructive comments. As you pointed out, we added a few sentences in this manuscript (third paragraph of Introduction in page 2, first paragraph of 2.4.2 section in page 6, and fifth paragraph of 3.2 section in page 10–11).

Reviewer 2 Report
The manuscript “A study on the behavior patterns of the liquid aerosols using disinfectant chloromethylisothiazolinone/methylisothiazoli-3 none solution” concerns the study of the behavioral characteristics of the substances methylisothiazolinone (MIT) and chloromethylisothiazolinone (CMIT)) when they convert to liquid aerosols. The topic of this work relates to human exposure to these substances, and it is of high interest since it concerns human health and safety. However, there are significant deficiencies of this study
- The novelty of this study and its contribution to literature is not clearly reported and justified.
- The usefulness of the outcome of the study is not identified.
- No recovery and precision data are presented for the whole protocol.
- The results are not compared or discussed with other studies on the same topic in literature like reference [26] or other studies that are not included in the references:
- Experimental determination of indoor air concentration of 5-chloro-2-methylisothiazol-3(2H)-one/ 2-methylisothiazol-3(2H)-one (CMIT/MIT) emitted by the use of humidifier disinfectant. Environ Anal Health Toxicol. 2020 Jun; 35(2): e2020008. Published online 2020 Jun 30. doi: 10.5620/eaht.e2020008
- Characteristics of Exposure to Chloromethylisothiazolinone (CMIT) and Methylisothiazolinone (MIT) among Humidifier Disinfectant-Associated Lung Injury (HDLI) Patients in South Korea. Molecules. 2020 Nov 12;25(22):5284. doi: 10.3390/molecules25225284.
- There is no discussion on the simulation of the experimental conditions applied and the real conditions that human exposure may occur.
- The parameter of time in the experiments has not been considered. For how long was the sample solutions aerolised?
- It appears that no real disinfectant samples (commercial products) were used in the study. Why not?
- They authors declare that the study investigated the behavioral characteristics however it is not clear which are the parameters that describe the term “behavioral characteristics”.
- The manuscript describes in detail the experimental part, however the number of Tables is big and some of them are redundant and could be included as supplementary material, as for example Table 3 which describes the preparation of the working and sample solutions.
Other minor comments
Line 40: The sentence “…that can be exposed to the human body through..” should be rephrased
Line 116: The abbreviations WS and SS should be explained in the first occurrence in the main text
Lines 117 - 121: it is not clear why a primary solution and a primary standard with similar purities were used.
Table 5. The actual concentrations range and number of calibration points should be reported and not codes of standards.
Author Response
Response to Reviewer 2 Comments
General Comments
The manuscript “A study on the behavior patterns of the liquid aerosols using disinfectant chloromethylisothiazolinone/methylisothiazoli-3 none solution” concerns the study of the behavioral characteristics of the substances methylisothiazolinone (MIT) and chloromethylisothiazolinone (CMIT)) when they convert to liquid aerosols. The topic of this work relates to human exposure to these substances, and it is of high interest since it concerns human health and safety. However, there are significant deficiencies of this study
I think the following should be done:
- The novelty of this study and its contribution to literature is not clearly reported and justified.
Response: Thank you for the comments. A few sentences have been added to the Introduction and Conclusions to further emphasize the novelty of this study (third paragraph of Introduction in page 3 and third paragraph of Conclusions in page 13).
- The usefulness of the outcome of the study is not identified.
Response: Thank you for the comments. The usefulness of the study results is added at the end of the conclusion (third paragraph of Conclusions in page 13).
- No recovery and precision data are presented for the whole protocol.
Response: The QA data for the analytical system that quantified MIT and CMIT are presented in Figure 2 (page 6). The QA data is presented in terms of relative standard error (RSD) and limit of detection (LOD), respectively.
The recovery of MIT and CMIT in liquid aerosols are presented in Figure 4 by calculating the mass of MIT and CMIT relative to the mass of the liquid aerosol as a concept of “relative concentration” (page 8).
- The results are not compared or discussed with other studies on the same topic in literature like reference [26] or other studies that are not included in the references:
- Experimental determination of indoor air concentration of 5-chloro-2-methylisothiazol-3(2H)-one/ 2-methylisothiazol-3(2H)-one (CMIT/MIT) emitted by the use of humidifier disinfectant. Environ Anal Health Toxicol. 2020 Jun; 35(2): e2020008. Published online 2020 Jun 30. doi: 10.5620/eaht.e2020008
- Characteristics of Exposure to Chloromethylisothiazolinone (CMIT) and Methylisothiazolinone (MIT) among Humidifier Disinfectant-Associated Lung Injury (HDLI) Patients in South Korea. Molecules. 2020 Nov 12;25(22):5284. doi: 10.3390/molecules25225284.
Response: The results of the previous research you mentioned above are briefly discussed in Introduction (second paragraph of Introduction in page 2).
- There is no discussion on the simulation of the experimental conditions applied and the real conditions that human exposure may occur.
Response: Thank you for the comments. The purpose of this study is to evaluate the behaviour patterns of liquid aerosols for accurate quantitative analysis of them. However, as the reviewer commented, we briefly mentioned it in the introduction (second paragraph of introduction in page 2).
- The parameter of time in the experiments has not been considered. For how long was the sample solutions aerolised?
Response: Thank you for the comments. The aerosols were generated for 2 hours. (first paragraph of 2.4.2 section in page 6.
- It appears that no real disinfectant samples (commercial products) were used in the study. Why not?
Response: This study is a basic study to accurately evaluate the concentration level of aerosols exposed under actual liquid chemical product usage conditions. The aerosol behavior was assessed by measuring and comparing the concentrations of liquid aerosol and its components under controlled aerosol exposure conditions (third paragraph of conclusions in page 13).
- They authors declare that the study investigated the behavioral characteristics however it is not clear which are the parameters that describe the term “behavioral characteristics”.
Response: We briefly mentioned the “behavioral characteristics” in third paragraph of Introduction in page 2.
- The manuscript describes in detail the experimental part, however the number of Tables is big and some of them are redundant and could be included as supplementary material, as for example Table 3 which describes the preparation of the working and sample solutions.
Response: Table 3 is moved to the supplementary material (Table S1).
Minor Comments
Point 1: Line 40: The sentence “…that can be exposed to the human body through..” should be rephrased.
Response 1: Thank you for pointing out. We rephrased the sentence (first paragraph of Introduction in page 1).
Point 2: Line 116: The abbreviations WS and SS should be explained in the first occurrence in the main text.
Response 2: Thank you for pointing out this error. We revised the sentence (first paragraph of 2.2 section in page 3). WS is abbreviation of working standard and SS is abbreviation of sample solution.
Point 3: Lines 117 - 121: it is not clear why a primary solution and a primary standard with similar purities were used.
Response 3: In the case of primary solution, in order to sufficiently generate the CMIT/MIT aerosols, a product that can be purchased in bulk was purchased as a primary solution (first paragraph of 2.2 section in page 3).
Point 4: Table 5. The actual concentrations range and number of calibration points should be reported and not codes of standards.
Response 4: Thank you for the suggestion. We reported the actual concentration ranges in Figure 2 (page 6) (According to the comments of other reviewers, Table 5 has been changed to Figure 2).
